# Involuntary Pregnancy Loss and Nursing Care: A Meta-Ethnography

**DOI:** 10.3390/ijerph17051486

**Published:** 2020-02-25

**Authors:** Sara Fernández-Basanta, María-Jesús Movilla-Fernández, Carmen Coronado, Haizea Llorente-García, Terese Bondas

**Affiliations:** 1Department of Health Sciences, Faculty of Nursing and Podiatry, University of A Coruña, Naturalista López Seoane s/n, 15471 Ferrol, Spain; maria.jesus.movilla@udc.es (M.-J.M.-F.); carmen.coronado@udc.es (C.C.); 2University Hospital Complex of Ferrol, Galician Health Service (SERGAS), Av. da Residencia, S/N, 15405 Ferrol, Spain; haizea.llorente@udc.es; 3Faculty of Health Sciences, University of Stavanger, PO Box 8600, Forus, 4036 Stavanger, Norway; terese.e.bondas@uis.no

**Keywords:** caring, meta-ethnography, meta-synthesis, midwives, miscarriage, nurses, perinatal loss, qualitative research, stillbirth

## Abstract

Healthcare professionals find the care of parents following an involuntary pregnancy loss stressful and challenging. They also feel unprepared to support bereaved parents. The challenging nature of this support may have a personal impact on health professionals and the care provided to parents. The aim of this meta-ethnography is to synthesise nurses’ and midwives’ experiences of caring for parents following an involuntary pregnancy loss. A meta-ethnography of ten studies from five countries was carried out. GRADE CERQual was assessed to show the degree of confidence in the review findings. An overarching metaphor, *caring in darkness*, accompanied by five major themes provided interpretive explanations about the experiences of nurses and midwives in caring for involuntary pregnancy losses: (1) Forces that turn off the light, (2) strength to go into darkness, (3) avoiding stumbling, (4) groping in darkness, and (5) wounded after dealing with darkness. Nursing staff dealt with organizational difficulties, which encouraged task-focused care and avoidance of encounters and emotional connection with parents. However, nurses and midwives might go beyond in their care when they had competencies, support, and a strong value base, despite the personal cost involved.

## 1. Introduction

Involuntary pregnancy loss is a relatively common occurrence. The exact prevalence of these losses is difficult to estimate. Previous research establishes that between 20–30% of pregnancies end in miscarriage worldwide. The precocity of the event or the intoxication of records with induced abortion rates could make it difficult to estimate miscarriages [1]. Regarding stillbirth, an estimated 2.6 million babies were stillborn worldwide in 2015 [2]. However, these statistics do not represent the totality of losses, since it is estimated that less than 5% of stillbirths have death records. Nevertheless, these numbers show that it is very likely that health professionals will encounter a significant number of families who have experienced a pregnancy loss in a current or previous pregnancy [3].

Involuntary pregnancy loss has been defined as the spontaneous demise of a pregnancy [4]. In this meta-ethnography, we included miscarriages and stillbirths. Miscarriage is an unplanned pregnancy loss before 20–24 completed weeks [5], and stillbirth is the death of a foetus that has reached a birth weight of 500 g, gestational age of 22 weeks, or crown-to-heel length of 25 cm [6].

### 1.1. Background

Pregnancy loss generates a varied, dynamic, and highly individualized response in parents [7,8]. Some parents may feel guilt and shame, others might feel relief and hopeful about the future, and others still may feel ambivalent about pregnancy and loss [9]. While there is a significant body of studies focused on the experience of heterosexual women who experience a pregnancy loss, the impact of miscarriage and stillbirth on male partners has been largely overlooked in academic research [10,11]. Emotional expressions are usually scarce and hushed, but the impact on their health is significant, especially if they do not have the opportunity to grieve openly [12].

Professional support for parents experiencing the involuntary pregnancy loss is needed, and nursing staff are those who accompany parents during pregnancy losses [13]. This care should go beyond medical cure and should be based on the conception of the person as an indivisible entity that includes body, soul, and spirit [14]. For these reasons, nurses and midwives are the reference professionals for parents [13].

In practice, professionals find the care of bereaved parents stressful and challenging. Therefore, this support could be superficial and focused on medical tasks rather than caring for the whole person [13]. The challenging nature of this support may have a personal impact on nurses and midwives. Health professionals may be required to set aside their own emotional responses and to focus on the tasks demanded by their work context and parents [15].

The care of involuntary pregnancy losses has evolved from a model of denial and protection to open support. Nevertheless, current care in pregnancy losses is inconsistent [13]. In miscarriages, the emergency department is, frequently, the only opportunity for parents to receive formal support [16]. In stillbirths, the guidelines mainly focus on medical management, while emotional support is relegated to the judgment of each healthcare professional [17]. Healthcare occurs mostly in hospital settings, and the requirements of this context influence the care parents receive [18].

### 1.2. Theoretical Perspective

Caritative Caring Theory [19] may serve to understand the complexity and wholeness of the experience of nurses and midwives regarding the care provided to pregnancy losses.

Ontologically, a human being is conceived as an indivisible entity that includes body, soul, and spirit. The human being has a unique vision of the world as a result of individual experiences and, at the same time, is connected with the culture of the person. According to Eriksson [20], caring maintains and enables health and well-being.

The purpose of caring is to alleviate suffering and to promote health and life. The ethos of care includes caritas, love and charity, the respect of health professionals for the dignity of the person, and a strive for genuine communion and understanding of the unique human being. Therefore, caritative care is based on the relationship between the person who needs and hopes for care and the person who is caring through a genuine communion and understanding for the unique human being [19].

Caring communion is understood as an act of human sharing in a caring relationship. Caring does not only imply performing professional nursing duties but a motive. Professional caritative caring is a genuine mature attitude of responsibility, courage, and wisdom. Caritative caring involves an encounter in which suffering and caring humans are participants in their own lived worlds of experiences and wishes [20].

Parents’ experience following an involuntary pregnancy loss has been addressed in the literature [21], but studies are scarce and there is a lack of theory from the nursing and midwifery care perspective. The aim of this meta-ethnography is to synthesise nurses’ and midwives’ experiences of caring for parents following an involuntary pregnancy loss.

## 2. Materials and Methods

Meta-ethnography is a method that involves knowledge synthesis to enrich human discourse by translating individual qualitative studies into one another, reinterpreting and transforming findings [22]. This study follows the seven phases of synthesis described by Noblit and Hare (1988) [22]: (1) getting started, (2) deciding what is relevant to the initial interest, (3) reading the studies, (4) determining how the studies are related, (5) translating the studies into one another, (6) synthesising translations, and (7) expressing the synthesis. This review has been written in accordance with the eMERGe meta-ethnography reporting guidance [23] (Table A1).

### 2.1. Search Methods

A comprehensive systematic search strategy was undertaken in the PubMed, Scopus, CINAHL, PsychINFO, and Web of Science databases in August 2019. To maximize coverage and to improve reliability [24], back-and-forth tracking and manual searches were conducted on the International Journal of Nursing Studies, Midwifery, Birth, Women and Birth, and Sexual & Reproductive Healthcare. The search was performed with no time limitations. The search strategy was constructed according to the phenomenon of interest (involuntary pregnancy loss), the purpose of the study or evaluation (care experiences), the sample (midwives and nurses), and the type of research (qualitative research). In each, search terms and medical subject headings were included. These terms were combined using the Boolean operators OR, AND, and NOT. Truncations were also employed to ensure a broad search.

Titles, abstracts, and full texts of original qualitative or mixed articles were examined, and those considered suitable according to the research objective were included. Inclusion was restricted to studies in which the sample comprised nursing staff and in which the type of loss was miscarriages and stillbirths. Papers not in English, Portuguese, or Spanish were excluded. The elaboration of search strategies was carried out by the first author.

### 2.2. Search Outcomes

Database searches yielded 742 records. Supplementary searches did not provide any additional records. The selection process of the articles began with the elimination of 281 duplicate articles. The titles and abstracts of 517 retrieved papers were assessed against the inclusion and exclusion criteria. Thirty-five articles were read in full and examined in relation to the inclusion and exclusion criteria. At this stage, 25 papers were excluded for sample reasons, interest phenomenon, type of loss, methodology, and type of paper. In both stages, the entire selection process was executed by S.F.-B. and H.L.-G., and in team sessions with M.-J.M.-F. and C.C., the authors reached consensus. The final sample was 10 articles (Figure 1).

### 2.3. Quality Appraisal

Each primary study was evaluated for quality using the Joanna Briggs Institute Qualitative Assessment and Review Instrument (QARI) [25]. Included articles were considered to have high quality with regarding their goals, designs, analyses, and results, providing useful knowledge on the topic (Table 1). The quality assessment was performed in team sessions by S.F.-B., H.L.-G., and M.-J.M.-F.

### 2.4. Data Extraction and Synthesis

Data extraction and synthesis was led by S.F.-B. Initial data extraction was carried out by S.F.-B. and subsequently discussed in team sessions. It involved the rereading of the included articles to describe each study’s aim, sample, method, type of loss, data collection methods, and key findings (Table 2) to provide context.

The primary articles were read and translated starting with the richest paper in terms of data [26,27] (step 3).

The first- and second-order [37] concepts were extracted across the full primary study by S.F.-B. and recorded in a Microsoft Word table. Using the constant comparison method (step 4), different concepts were compared by S.F.-B., T.B., and M.-J.M.-F. in search of similarities and contrasts, which led to the formation of new concepts and to the adoption of existing concepts. This was done by systematically and sequentially comparing concepts using the registered study characteristics (Table 2).

In step 5, the concepts were organized by S.F.-B. in conceptual piles and then the piles were discussed and reorganized by all the authors. It was determined that the studies met the criteria for reciprocal translation, so the first- and second-order constructions were placed, allowing the development of third-order constructions [37]. These new understandings were added to the reciprocal synthesis, building on the themes and metaphor (step 6). Five themes were developed in relation to preserving the studies’ key contents. All the authors agreed on the metaphoric themes and subthemes and on the overarching metaphor.

The Confidence in the Evidence from Reviews of Qualitative research (CERQual) approach was used to show the degree of confidence in the review findings [38] (Table A2).

## 3. Results

The sample consisted of 10 qualitative primary articles. These studies were conducted in Ireland, the United States of America, Canada, Spain, and New Zealand; include a total of 129 nurses and midwives, predominantly in the hospital setting; and focused on stillbirth. Descriptive designs were more common than interpretive research designs. Semi-structured and in-depth interviews and focus groups were used to collect the data (Table 2).

A reciprocal synthesis, using the metaphor *caring in darkness* (Figure 2), provided interpretive explanations of the experiences of nurses and midwives in the care of involuntary pregnancy losses. This metaphor, accompanied by 5 themes, symbolizes the care experience of nurses and midwives in involuntary pregnancy losses. Unsupportive organizational culture, lack of preparation or knowledge, and an emotionally demanding care represented metaphorically by forces that turn off the light hindered the care. *Darkness* also represents the emotional state of parents following a pregnancy loss.

Going into darkness caused fear because healthcare professionals feel unprepared or unsupported, because they did not want to hurt parents even more, and because of the emotional implication involved. Despite that, having leader and peer support and guidance, previous experiences, and a strong value base encouraged nurses to care and to have the strength to go into darkness.

However, their motive to care may not be enough for many of them and they decided to avoid stumbling. Instead, the care provided was task-focused, in which encounters and emotional connections with parents were avoided. Those who went beyond the tasks tried to care in the best possible way despite the difficulties, groping in darkness. Wounded after dealing with darkness illustrates the consequences on nurses and midwives.

The CERQual assessment showed high confidence in all the themes, meaning it is likely that they reasonably represent nurses and midwives’ experiences in the care of involuntary pregnancy loss (Table A2).

### 3.1. “Forces that Turn Off the Light”

Unsupportive organizational culture, lack of knowledge and preparation, and emotionally demanding care were difficulties identified by nurses and midwives in the care of involuntary pregnancy losses. These were, metaphorically, forces that turned off the light and that left them in darkness.

#### 3.1.1. Unsupportive Organizational Culture

Nurses and midwives found the health system unhelpful when it came to handling their own grief. This lack of backing was further exacerbated in early pregnancy losses [26,27,28,29]. A midwife referred to the abandonment of the health system as follows: *“Sometimes, you are left to deal with it on your own; especially, sometimes, it’s on nights.… By the time you come back, sometimes, it just doesn’t really emerge again.”* Midwife [26,27].

Cost-efficiency policies may have contributed to nurses minimizing the emotional aspects of care to focus on physical aspects [30]. Beaudoin and Ouellet (2018) [31] reported that nurses sometimes had to carry out medical tasks that deprived parents of having moments of intimacy: *“Yes, (for) staying in the room with the parents, some are comfortable with that, but sometimes, they would rather be by themselves. I think that that can be harmful.”*

The non-provision of whole care could also depend on the clinical setting where women were hospitalized. In the gynaecology service, mainly aimed at surgical pathologies, there was a tendency to marginalize the emotional aspects of work and to focus on the intervention treatment [30]: *“It made you come back to it, really think about what these women are going through, rather than just treating it as evacuation of uterus. We were a gynae unit on its own, and now, we’re up here as a surgical on gynae unit. We have a few staff on the ward that would be surgically trained. They would have found it difficult at the beginning: some days you were busy and you didn’t have time for people; they were just rushed in and rushed out again.”* Nurse.

Nurses and midwives reported that the workload took time from talking and being with parents, complicating the establishment of a bond of trust with them [26,27,31]. Moreover, they could also be involved in the care of families with pregnancy loss while others simultaneously could not [26,27,28,32,33]. This situation required an emotional and behavioural adjustment with respect to listening, physical presence, and handling privacy and intimacy on the way from one room to another, which was hard for them [26,27,28,34]. *“The worst is having a labour patient at the same time, if you have a demise: being happy in one room and sad in another and you don’t want to get the rooms wrong, and it is really hard.”* Nurse [34]. Hutti et al. (2016) [32] reported that the development of that ability to adapt their emotional switches from care of the loss to another situation not so emotionally demanding depended on the idiosyncrasy of the services, being more difficult for delivery nurses due to such opposite emotions.

On the other hand, most services lacked exclusive spaces to take care of the loss. This situation favoured the contact of families with pregnancy losses with others who had not suffered any losses [28,31]. Furthermore, in early pregnancy losses, the lack of respectful places to leave the dead baby generated an emotional dilemma in nurses [30,31]. *“For my part, I would have liked… if there was a place for that instead of… going to weigh it in the same place as the placentas.”* Nurse [31].

Administrative paperwork management, such as the baby name registration, autopsy, spiritual care provider, and emergency baptism, was also reported as a burden [31,33]. *“Some nurses don’t mind taking care of patients with a loss; it’s the paperwork that is so daunting!”* Nurse [33].

#### 3.1.2. Lack of Preparation and Knowledge

Midwives and nurses reported a lack of preparation and knowledge regarding how to care for these parents, how to communicate with them, and how to behave besides a scarcity of tools to manage the emotional demands of parents and their own [26,27,28,29,30,31,32,33,34,35]. This caused them to feel insecure when caring [32].


*“When I started working in the operating room, I was just put into a (perinatal loss) case, and I didn’t know what to expect. I never really had any training on how to deal with the patient or with my own feelings. Unfortunately, sometimes, you are just thrown into things and you just learn.”*
Nurse [32].

Some of them reported a lack of appropriate communication skills, and they found it difficult to know what, when, and how much to say; how to approach families; and how much to get involved [26,27,28,29,30,31,33,34,35,36]. *“My personal difficulty is this: what do you say to this woman? To me, everything seems banal.”* [34].

A particularly controversial aspect was the lack of congruence regarding socially accepted behaviour with parents. This caused doubts about whether to show or contain their own emotions [26,27,28,29,30,31,33,35]. While some understood that emotional expression is a tool for establishing a bond of trust with the family, others felt that the emotional demonstration could further harm the parents, since the loss belonged to the parents [26,27,29,30,31,33,35].

Other situations that challenged nurses and midwives were communication with parents with recurring early losses, the inability to answer conclusively to parents’ questions, the provision of information regarding practical aspects such as funeral arrangement, what happens next or who to contact, and the management of unexpected or unfamiliar behaviours or signs of exacerbated pain and suffering [26,27,28,31,32].


*“They really don’t recognize at all that they have had a baby, and they don’t want anything to do with either the funeral arrangements or anything like that.… To accept that their way of dealing is also normal and natural for them, there can sometimes be a bit of conflict there.”*
Midwife [26,27].

Unsupportive resources, training, or learning opportunities during the student stage and later as professionals were reported as caused by a lack of knowledge [28,29,30,31,33,34]. *“We have been taught a lot of techniques—a lot of theory—had a lot of practice of how to place such-and-such apparatus,… but nobody has taught us, or me personally during my career: nobody has taught me how to confront these cases and how I can help these people.”* Midwife [34]. In the study of McCreight (2005) [30], carried out in the gynaecology service, some nurses noted that the contents of the training were incomplete by not meeting the emotional demands of the care.

Consequently, this led nurses and midwives to learn and care ad hoc, where their own personal or work experiences provided learning for the care of future losses [28,30,32]. Peer learning from experienced colleagues was also another resource [30], as a nurse reported: *“The training you get on the ward here is your own way of dealing with any type of bereavement. Plus, there are very experienced staff, and basically what you do is you learn from them, you take away parts of their way of dealing with it and adopt it for your own use; that’s basically how the training is done, nothing formal as such.”*

#### 3.1.3. Emotionally Demanding Care

The majority of nurses and midwives experienced care as hard, difficult, and even a failure in the care process [29,32,33,34,36]. Some of them approached care from the weight of responsibility, in which their actions would have an impact on the well-being of parents [32,36]. *“It’s scary to know that they’re not going to forget anything that is said that day; so, be the right person at the right time for them. So, it’s a huge responsibility, but it is a great honor.”* Nurse [36]. Furthermore, some of them may feel unprepared to witness the emotional grief and suffering of parents and even the delivery [29,31,32].


*“The first time I could see the parts of the baby. I was never really prepared for that. It was just shocking, and there was a moment after doing that case when I almost wanted to speak to a manager and say, ‘I may not be able to do these types of cases.’”*
Nurse [32].

Apart from managing and dealing with parents’ feelings, the care required emotional involvement on their behalf, as this midwife highlighted: *“The fact is it’s hard for you, for you not as a midwife, as a person.”* [34]. This deep and emotional implication had consequently a personal cost, especially when personal traumas were not overcome, when their beliefs and values were contrary, or depending on the relationship established with the parents and the story behind them, for instance, multiple pregnancy losses [26,27,28,29,30,31,32,34,36].


*“It’s very intense for the midwife looking after these people. There have been days when I’ve gone home and cried over different situations from here.”*
Midwife [29].

### 3.2. “Strength to Go into Darkness”

Leader and peer support and guidance, having professional and personal experiences, and their convictions regarding their role were reported as motivating elements to deal with care in involuntary pregnancy losses. These created strengths to go into the darkness.

#### 3.2.1. Leader and Peer Support and Guidance

Peer support provided security to care for involuntary pregnancy losses [31,32]. Moreover, a nurse from the Beaudoin and Ouellet (2018) [31] study underlined the key role that nursing leaders have: *“I think so, and that, I find that very brilliant: the respect that the assistants have toward their colleagues. The assistants are very much looking out for and listening to their gang.”* Leader and peer supervision and guidance, and training were highly demanded by nurses and midwives, since the knowledge about how to care for the loss provided them with comfort, confidence, and reduction of concerns [26,27,28,30,31,32,33,34,36]. Even nonspecific training was considered beneficial [29].

#### 3.2.2. Professional and Personal Experiences

They were considered as a source of learning and improvement in care [26,27,28,30,31,33]. Roehrs et al. (2008) [33] highlighted the enrichment of having those experiences in the student stage despite the hardness of the losses: *“[I had a] great orientation. I had 2 or 3 full-term losses in my orientation, and they were very difficult situations. I read a lot of info, talked with [others,] and reviewed reference books.”* Nurse. Specifically, the nurse–parent bond was strengthened when nurses and midwives had personal experiences of loss or children, since these provided understanding and knowledge with which to empathize and feel safe [26,27,28,30,31,33]. *“I feel comfortable taking care of patients with early losses because that is where my personal experience lies. I find it more difficult to take care of patients with full-term demises.”* Nurse [33].

#### 3.2.3. Convictions about Care

Nurses and midwives recognized themselves as key in the provision of care and that they should be strong, supportive, and present for parents despite the personal cost involved [29,31,33,34], as a nurse reported: *“You don’t want that feeling of having that pain and anguish for somebody because you do, but the patient needs care and you need to do it.”* [29]. In addition, they considered themselves the ones providing compassion, understanding, and support [34,36]: *“Well, perhaps the midwife will be the one who has to attend this confinement… because it is supposed that we are the ones who are prepared and will be with and support them during this procedure.”* Midwife [34].

On the other hand, the care of involuntary pregnancy losses was significant for them, since the care was both challenging and difficult besides meaningful [30,31,32,33,35,36]. The idea of making the loss bearable caused them to feel reward, gratitude, usefulness, and honour [33,36]: *“Well it’s really—it’s very difficult, but it’s probably in some ways—it’s more rewarding to help a family through a crisis like this,”* Nurse [36].

### 3.3. “Avoiding Stumbling”

To avoid stumbling in darkness, nurses and midwives did not go beyond, provided a task-focused care, and avoided encounters and emotional connections with parents.

#### 3.3.1. Convictions about Care

Mainly, the care prioritized procedures over emotional aspects [31,32]. For instance, in early losses, the care becomes routine and the emotional aspect tends to disappear [28]: *“You’re kind of saying the same things. It’s very kind of routine, very much like a conveyer belt.”* Midwife. Particularly, Hutti et al. (2016) [32] showed that surgery nurses were distressed when mothers woke up crying. They tried to control their crying by sedating them: *“She had had Versed. We like to give the Versed because it does help when they’re crying. Some patients, even when they don’t have a loss, they’ll cry from the anaesthesia. So, we do give that.”*

#### 3.3.2. Avoiding Encounters with Parents

The lack of time due to high workload hindered establishing a bond of trust with parents, and therefore, care could have been superficial and without follow-up [26,27,31]. Moreover, Roehrs et al. (2008) [33] reported uncaring situations due to the impossibility of exclusive dedication to the loss: *“She had delivered in the bathroom. I felt bad that I was not in there with her when that happened, that I wasn’t there even though I ran in the door as soon as she called.”* Nurse.

Delegating care to more experienced colleagues was another mechanism to avoid encounters with parents, as a midwife said: *“A baby comes back from autopsy, and everybody runs in every direction. They don’t want to bath it or dress it or get it ready to put it in the little basket, and very often, the same one or two people are there.”* Midwife [26,27].

#### 3.3.3. Avoiding Emotional Connection

Nurses and midwives did not get emotionally involved with parents to avoid personal suffering and because they felt insecure and unprepared [28,31]. For example, some nurses put aside their feelings, others were able to depersonalize the baby to minimize the emotional burden of this loss, and even some of them had a feeling of rejection of care [26,27,28,29,31,32,33,34,35]. *“I think that sometimes I go in auto mode as I care for patients and don’t realize the emotional toll until I go home.”* Nurse [33].

### 3.4. “Groping in Darkness”

Nurses and midwives who went beyond the task-focused care did it in the best way they knew, which is represented by groping in darkness.

Midwives and nurses tried to provide care in the best possible way, despite the difficulties encountered [32,33,35], as this nurse reported: *“You have to just hope you made an experience OK for them, that you helped them in some way, because if you go home every day and think about all these people that have had horrible losses, you’re not going to care for the next one effectively.”* [32].

Care was based on the physical presence, on encouraging the expression of feelings of parents and normalizing the feelings of mothers, on comforting them physically and emotionally, and on the acknowledgement and memories created of the baby [26,27,29,31,32,33,34,36]. They also tried to individualize and adjust their care according to the situation, their relationships with patients, and the time available for care [32].

### 3.5. “Wounded after Dealing with Darkness”

The care of involuntary pregnancy losses caused a personal cost to nurses and midwives. This personal cost was the wounds caused by dealing with darkness.

On a personal level, nurses and midwives felt anxious, sad, grieved, anger, and inconsolable [28,29,31,32,36]. Sometimes, the feelings were so intense that they had an impact on their daily life [28,29], as this nurse expressed: *“I think more like exhaustion, like physical and mental exhaustion, is the best way to describe it: depressed. I could not sleep. I did not sleep well… hearing wailing, seeing the looks on their faces; you know, you just replay it over and over.”* [29].

In situations of high emotional demand, such as the care of several losses in one day or the simultaneous care of a pregnancy loss and healthy pregnancy, nurses and midwives felt emotionally drained and could trigger burn out [26,27,29,30,32,36]. *“It’s emotionally draining, not so much physically but emotionally. It’s hard to be taking care of a patient with a stillbirth and then have to go into a patient’s (room) who is delivering a live baby.”* Nurse [32].

Midwives even felt guilty and responsible for the death of the baby despite the situation being beyond their control [26,27,35]: *“I felt that I had missed something and that I should have been able to make that baby come alive. I don’t know why; it’s just, I have live bubbies not dead ones. Why did it happen…? What did I do wrong…? What didn’t I see happening…? Yeah… all those sorts of things. It was really raw emotions. I think you blame yourself for something like this, more than you think you do.”* Midwife [35]. Frustration at the lack of institutional support and non-satisfaction of care expectations was also reported [26,27,28,29].

The management of these feelings; the high physical, mental, emotional, and spiritual needs of the parents; and their lack of training caused overwhelming feelings, and some of them could feel helpless and unqualified [29,31,32,35]. *“It’s definitely overwhelming… When I started my orientation, we had plenty of classes—a newborn class, a postpartum class—but there was never much talk about (perinatal loss).”* Nurse [29]. In the Roehrs et al. (2008) study [33], some nurses came to consider leaving the service because of the feeling of guilt and not feeling able to properly care for these parents: *“The fear of personal wrong doing or what part of this could be or is my fault make it difficult to come back to work after you have cared for the family in labor”.*

## 4. Discussion

From the analysis of the 10 qualitative primary articles emerged the metaphor *caring in darkness* (Figure 2). This metaphor represents the nurses’ and midwives’ experience in the care of involuntary pregnancy losses. Darkness represents both the suffering of parents and the challenges when caring for parents. They had to deal with an unsupportive organizational culture, lack of preparation or knowledge, and an emotionally demanding care, while they found motives to get into the parents’ grief and suffering. The result of this confrontation was uncaring or was care based on lack of preparation and fear of hurting and on personal cost. This metaphor can be an incentive for action and could be useful for managers, leaders, and professionals, metaphorically providing candles that illuminate and guide nurses and midwives in the care of involuntary pregnancy losses.

Eriksson [19,20] states that the basic motive of caring is the caritas motive, where caritas is defined as altruistic love expressed in action. The readiness of nurses and midwives to share the parents’ struggle of suffering is an essential aspect of caritative caring. In a context of scarcity of economic resources and marked masculine technological thinking, nurses and midwives may feel insecure when providing caritative care, for fear of being labelled as weak or unprofessional. From a clinical point of view, caritas in western health systems means going beyond a role. From the nurse, this requires competence, motivation, and moral integrity [39]. Our findings showed that, apart from competencies and support, it was ethos, or human value base, that motivated care in involuntary pregnancy losses.

Nurses confront human suffering on a daily basis and are expected to provide genuine care to alleviate people’s distress rather than simple task-oriented responses [19]. Nurses and midwives are fundamental persons in the care of involuntary pregnancy losses and are better placed to build a genuine communion [14,19]. However, our results show that the care was mostly based on the performance of nursing tasks [28,29,30,31,32]. Nurses and midwives failed to establish a genuine communion with parents on many occasions.

A market mentality in health management systems may encourage the technical care and nonemotional involvement of nurses and midwives with parents [40]. This biomedical predominance is more pronounced in services such as gynaecology, emergencies, or surgery, where a wide variety of processes far from pregnancy losses are attended to. Therefore, the care could mainly address the accomplishment of medical tasks or may involve the medicalization of emotional aspects [30]. In this sense, Maturo (2012) [41] stated that the processes related to mental health are widely medicalized.

This context hinders the promotion of a caring culture and generates conflicts in health professionals, realizing that the interests of the administration are not equal to theirs. The Caritative Leadership theory [42] was created as a result of that and establishes that the entire caring culture and the leader are responsible for what is happening in the organization, especially in situations of noncaring, uncaring, nonchalance, and provocation. Leadership is understood as a powerful and fundamental tool in the delineation and maintenance of the ethical value base, direction, and content of nursing care in the complex demands of evidence-based, efficient, and cost-effective nursing care [43].

On the other hand, the majority of nurses see emotional engagement as a requirement of excellence in nursing practice [44]. The benefits of establishing a trust relationship with patients was reported in a recent meta-ethnography [45]. This relationship is a useful tool for personal communication sharing, which allows holistic nursing care, as it strengthens nurses’ ability to recognize and respond properly to a variety of unvoiced needs. The development and maintenance of this trust relationship with parents implies work by nurses and midwives, which adds to the invisible and emotional labour that is generally not recognized in nursing [46].

According to Hochschild (1983) [47], emotional labour refers to a worker’s endeavour to display emotions according to embedded social and cultural norms rather than what health professional actually feels. Emotional labour may be viewed as a western construction within Descartes’ tradition of separating the rational mind from the emotional body [48]. However, caring implies feeling and feeling involves personal vulnerability [44]. Phillips (1996) [49] showed that emotional labour is denigrated by its association with femininity and that it occupies a second level with respect to cognitive or technical abilities.

Our results informed that difficulties in care encourage avoidance behaviours and that nurses and midwives experienced emotional exhaustion when they were emotionally involved with parents [26,27,29,32,36]. The fact that emotional labour is not recognized and is thus undervalued by the majority of healthcare organizations may contribute to understanding the empirical link between emotional labour, emotional exhaustion, and professional burnout [47]. In this sense, clinical supervision, understood as interprofessional support and as guidance and reflection, has been recognized by the literature as useful for improving professional growth and collegiality and for preventing burnout that risks the health of the nurse and nursing care [50].

### 4.1. Strengths and Limitations

The use of meta-synthesis has increased in the last decade. Meta-ethnography is a distinct, complex, and increasingly common and influential qualitative methodology in health and social care research [23]. Especially in nursing, qualitative syntheses are considered a useful method for examining participants’ meanings, experiences, and perspectives, both deeply and broadly. Its employment is useful to identify research gaps; to inform the development of primary studies; and to provide evidence for the development, implementation, and evaluation of health interventions. This methodology involves a conceptual development that implies a fresh contribution to the literature, beyond the narrative and systematic literature reviews [51]. The elaboration of this meta-ethnography has followed the eMERGe reporting guidance [23]. The utilization of this guidance improves the transparency and completeness of the research process and, therefore, the quality of the meta-ethnography. This allows our results to contribute to the formation of robust evidence that serves as the basis for political and practical decision making.

On the other hand, a comprehensive search strategy has been carried out with the possibility of including articles in English, Spanish, and Portuguese, although no results were found in these last two languages. This search has been executed in two moments: an initial that was limited to the bibliography of the last ten years, and the definitive one without time limits. This has allowed a double check of the existing literature.

Another strength is that the studies were evaluated using the QARI criteria [25] and that the review findings were assessed with CERQual [38], confirming their transparency and reliability. This process improves the trustworthiness and applicability of the results in the clinical setting, decision-making, and future research.

Regarding the limitations, these are present in the composition of the sample. On the one hand, the cultural contexts of primary articles are located exclusively in western countries. On the other hand, the sample of the primary articles is almost exclusively feminine and mostly belonging to the hospital environment. However, nursing staff in primary care settings accompany these families after the loss and for a longer time, and in many cases, the bond created with the parents is stronger. New empirical research that addresses these limitations is needed.

### 4.2. Relevance to Clinical Practice

This meta-ethnography informs the body of knowledge in nursing science and enhances a change in clinical practice, since the results show the complexity of the care experience of nursing and midwifery in involuntary pregnancy losses.

Nurses and midwives require an organizational culture that is supportive for the development of a caring culture. At the formative level, the focus should be on the provision of whole care and on the establishment of genuine communion with parents. Therefore, training should be aimed at developing and strengthening skills that favour the connection with parents and at the provision of tools for managing the emotional demands of caring. Nursing leaders, due to their position among administration, nursing staff, and the proximity to care, are key to favouring the interconnection and the construction of bridges between them.

To expand knowledge of the care experience in involuntary pregnancy losses, further research focused on primary care midwives, who in many cases have the most lasting contact with parents, is required. Besides, it would be beneficial to know the personal experience of nurses and midwives in caring for these losses. This perspective would provide us with valuable information for practice and the education of future nursing professionals.

## 5. Conclusions

The overarching metaphor, caring in darkness, symbolises the experiences of nurses and midwives in the care of involuntary pregnancy losses. Darkness represents both the suffering of parents and the challenges when caring for parents. Nurses and midwives dealt with organizational difficulties, lack of knowledge, and with care that requires going beyond. This context encouraged task-focused care and avoidance of encounters and emotional connection with parents. Metaphorically, they avoided stumbling in the dark. However, some nurses and midwives went beyond in their care when they had competencies, support, and a strong value base despite the subsequent wounds involved in dealing with darkness. Those who went beyond the tasks tried to care in the best possible way, symbolized through the theme groping in darkness. Deepening the personal experience of nurses and midwives who care for pregnancy losses would complement our results and allow to have a complete overview of this experience. These results could improve knowledge in nursing science and could encourage change in clinical practice.

## Figures and Tables

**Figure 1 ijerph-17-01486-f001:**
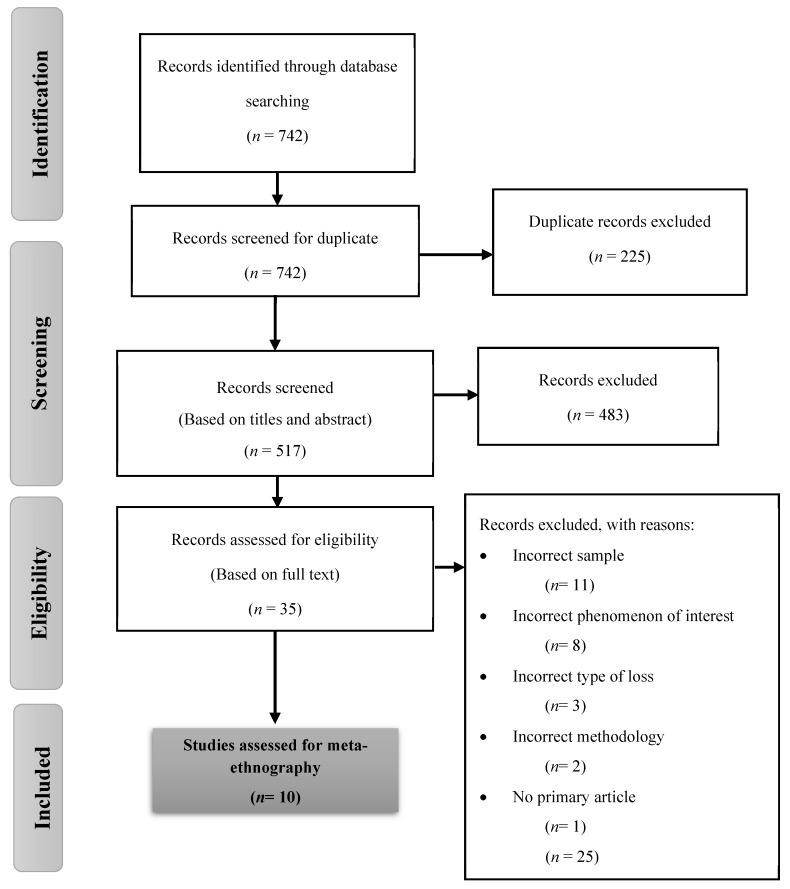
PRISMA flowchart.

**Figure 2 ijerph-17-01486-f002:**
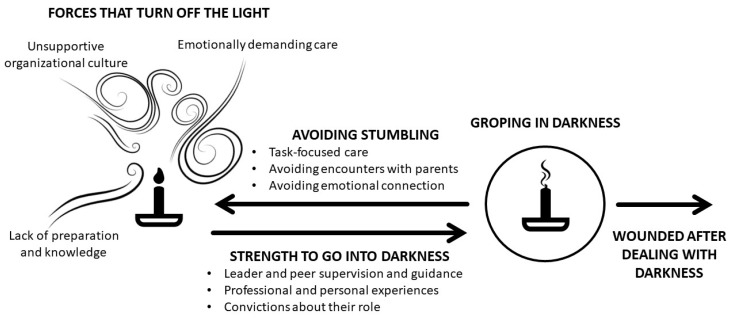
Caring in darkness.

**Table 1 ijerph-17-01486-t001:** Quality assessment of included studies [25].

Article	Questions									
	1	2	3	4	5	6	7	8	9	10
Nallen (2006, 2007) † [26,27]	**✓**	**✓**	**✓**	**✓**	**✓**	**✓**	–	**✓**	**✓**	**✓**
Nash (2018) [28]	**✓**	**✓**	**✓**	**✓**	**✓**	**✓**	–	**✓**	**✓**	**✓**
Willis (2019) [29]	**🗴**	**✓**	**✓**	**✓**	**✓**	**✓**	**🗴**	**✓**	–	**✓**
McCreight (2005) [30]	**✓**	**✓**	**✓**	–	**✓**	**✓**	**🗴**	**✓**	**✓**	**✓**
Beaudoin and Ouellet (2018) [31]	**✓**	**✓**	**✓**	**✓**	**✓**	**✓**	**🗴**	**✓**	**✓**	**✓**
Hutti (2016) [32]	–	**✓**	**✓**	**✓**	**✓**	**✓**	**✓**	**✓**	**✓**	**✓**
Roehrs et al. (2008) [33]	**✓**	**✓**	**✓**	**✓**	**✓**	**✓**	**✓**	**✓**	**✓**	**✓**
Martinez-Serrano et al. (2018) [34]	**✓**	**✓**	**✓**	**✓**	**✓**	**✓**	–	**✓**	**✓**	**✓**
Jones and Smythe (2015) [35]	**✓**	**✓**	**✓**	**✓**	**✓**	**✓**	**✓**	**✓**	**✓**	**✓**
Jonas-Simpson et al. (2010) [36]	**✓**	**✓**	**✓**	**✓**	**✓**	**✓**	–	**✓**	**✓**	**✓**

Abbreviations: **✓** yes—unclear **🗴** no; critical appraisal questions: (1) Is there congruity between the stated philosophical perspective and the research methodology? (2) Is there congruity between the research methodology and the research question or objectives? (3) Is there congruity between the research methodology and the methods used to collect data? (4) Is there congruity between the research methodology and the representation and analysis of data? (5) Is there congruity between the research methodology and the interpretation of results? (6) Is there a statement locating the researcher culturally or theoretically? (7) Is the influence of the researcher on the research, and vice-versa, addressed? (8) Are participants and their voices adequately represented? (9) Is the research ethical according to current criteria, or for recent studies, is there evidence of ethical approval by an appropriate body? (10) Do the conclusions drawn in the research report flow from the analysis or interpretation of the data? ^†^ Two parts of the same study.

**Table 2 ijerph-17-01486-t002:** Characteristics of included studies.

Authors (Year) Location	Methodology	Aim	Sample and Setting	Type of Loss	Data Collection Methods	Key Findings
Nallen (2006, 2007) Ireland [26,27]	Descriptive qualitative methodology	To explore midwives’ views regarding the provision of bereavement support to parents affected by perinatal death	18 hospital midwives	Perinatal death	3 focus groups	The findings centred on 5 major themes which emerged from the data: role recognition, prerequisites to bereavement support, perceived barriers to bereavement support, coping strategies, and spiritual support.
Nash et al. (2018) Ireland [28]	Descriptive qualitative design	To explore the perceptions of midwives caring for women experiencing early pregnancy loss.	8 midwives *(maternity hospital)*	Early pregnancy loss *(<13 gestational week)*	Face-to-face semi-structured interviews	Themes identified were: “coping with the experience of early pregnancy loss”, “compassionate care for women and midwives”, and “what midwives found difficult”.
Willis (2019) USA [29]	Descriptive qualitative methodology	To describe the experience of caring for women with a perinatal loss from the perspective of the nurse and to determine the extent to which the response to perinatal loss reflects a process.	9 labour and delivery nurses	Perinatal loss *(>20th gestational week to 1-month post birth)*	In-depth interviews	Several themes depicting nurses’ experience were identified: struggling with emotions, carrying on in the moment, being present for the patient, expressing conflict, and taking care of self. A process was identified by nurses describing their response to perinatal loss. The process began with recognition of the loss and progressed through phases including the recognition of their emotional impact, connecting with the mother, dealing with emotions, acting professionally, preparing to return to work, and never forgetting the woman.
McCreight (2005) Ireland [30]	Not mentioned	To collaboratively explore with gynae nurses how they constructed meanings through their narratives in relation to the professionally defined but personally experienced events of pregnancy loss.	14 gynaecological nurses	Pregnancy loss	Semi-structured in-depth interviews	Emotion can be conceived of as a valid resource for professionals when integrated into a nurse’s matrix of professional understandings. The study also demonstrates that value should be attached to emotional work which may not be fully visible, particularly for nurses working in gynaecological units. The emotional needs of nurses need to be fully acknowledged through recognition of the importance of managed emotion in the construction of professional knowledge.
Beaudoin and Ouellet (2018), Canada [31]	Fourth-generation constructivist evaluative method	To explore the factors influencing the practice of nurses with families experiencing perinatal loss in a secondary obstetric care centre in the Quebec region	7 obstetric nurses 3 managing nurses	Perinatal loss *(Death of a baby during the pregnancy (>20 gestational week) or a few days to a few weeks after childbirth)*	Semi-structured interviews	Five themes were identified: the quality of the relationship between the nurse and the bereaved family, the nurse’s personal characteristics, the emotions felt by the nurse, work organization on the hospital unit, and the context in which nursing care is provided to families. These themes draw attention to the importance of building a solid relationship of trust with bereaved families in which honesty, empathy, human warmth, and listening have a central place.
Hutti (2016) USA [32]	Not mentioned	To examine the experiences of, meaning for, and personal consequences for obstetric, emergency, and surgical nurses caring for women after foetal death and to determine how these nurses use Swanson’s caring processes in providing such care.	28 obstetric, surgery, and emergency nurses	Foetal loss *(Included miscarriages (<20 gestational week) and stillbirths (from >20 gestational week to birth))*	Focus group	Swanson’s caring processes were used as a way to describe the unified experiences of nurses who care for families after a perinatal loss. All nurses, regardless of specialty, used Swanson’s caring processes, but they used them preferentially according to situational exigencies and level of rapport developed with each patient.
Roehrs et al. (2008) USA [33]	Descriptive qualitative methodology	To describe support needs and comfort level of labour nurses caring for families experiencing perinatal loss.	10 labour nurses	Perinatal loss *(between >20 gestational week and 7 days old)*	Online surveys and follow-up interviews	Nurses are generally comfortable but find it difficult to provide perinatal bereavement care. Strategies for coping include focusing on needed care, talking to nursing peers, and spending time with their own family members. Nurses take turns providing care depending on ‘‘who is best able to handle it that day’’ and prefer not to be assigned a labouring patient in addition to the grieving parents. Developing clinical expertise is necessary to gain the comfort level and the skills necessary to care for these vulnerable families. Orientation experiences and nursing staff debriefing would help.
Martinez-Serrano et al. (2018) Spain [34]	Hermeneutic-interpretative phenomenological approach	To explore the experiences of midwives regarding the attention given during labour in late foetal death.	17 hospital midwives and 1 primary health centre midwife	Late foetal death *(≥1000 g birth weight, ≥28* gestational week *and ≥35 cm body length)*	3 focus groups	Two main themes were identified: professionals for life not death and organizing the work without guidelines. Midwives felt that there is a lack of social awareness related to the possibility of antepartum death that keeps the mourning hidden and affects the midwives’ practice during the late foetal death process. Midwives recognize difficulties in coping with a process that ends in death: organizations are not prepared for these events (not suitable rooms), and there is lack of training to cope with them and lack of continuity in the attention received by the parents when they are discharged.
Jones (2015) New Zeeland [35]	Hermeneutic interpretive phenomenology	To explore, understand, and appreciate the lived experience of midwives who have cared for parents whose baby has been stillborn.	5 self-employed midwives	Stillbirth *(the death of a baby before or during birth, from the 20 gestational week onwards, or weighing 400 grams or more at birth)*	Individual interviews	Two themes were identified. This paper focused on the theme “a pocketful of grief” which is made up of three subthemes: “shockwave”, “self-protection”, and “blameworthiness”. The death of a baby is a significant event for the midwife providing care.
Jonas-Simpson et al. (2010) Canada [36]	Exploratory qualitative descriptive method	To explore their experience of caring for families whose babies were born still or who died shortly after birth.	9 obstetrical nurses	Stillbirth *(>20 gestational week)*	In-depth face-to-face structured interview	Findings revealed that caring for bereaved families is a difficult yet meaningful experience valued by the nurses in this study. Connecting and supporting bereaved families with their babies was identified as an essential part of practice. Understanding from colleagues as well as time and space for reflection were helpful. Nurses offered mothers anticipatory guidance and described thinking about the mothers, even years later.

**Abbreviations**: United States of America (USA).

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
