# Peer review of "Involuntary Pregnancy Loss and Nursing Care: A Meta-Ethnography"

_ijerph, 2020, doi:10.3390/ijerph17051486_

Round 1
Reviewer 1 Report
If the corresponding university has removed the student's TFG from its repository. I accept the study because it is original and makes contributions for clinical practice. The authors should be more concise when explaining, in the conclusions, the metaphor " caring in the dark". They should better explain, in the conclusions, the five main topics.Author Response
Dear reviewer 1,
Thank you for your comment and appreciation of the manuscript ijerph-704190_Involuntary pregnancy loss and nursing care: A meta-ethnography. The conclusions section has been modified, expanded the explanation of the metaphor and the five main themes.
Reviewer 2 Report
Dear authors,
Thank you for the opportunity to read your interesting study, which highlights a very important topic.
Some reflections on your study, organized according to the headings.
Title:
Reflect the content of the study in a good way.
Abstract:
Overall well-written abstract describing the study in a good way.
I suggest that you have the same purpose in abstract and in the article, now they are different. My recommendation-include following aim in the abstract.
“The aim of this meta-ethnography is to synthesize nurses’ and midwives’ experiences of caring for parents following an involuntary pregnancy loss.”
I am critical that you present the major themes first- I think it is better to have the overarching metaphor first.
Introduction/Background:
Overall well-written and introduce me as a reader in a proper way. Relevant references.
The first paragraph is good in the introduction, but my recommendation is to start your introduction with line 43, about the prevalence and previous research, and after that include the first sentence (line 39) and the definition.
The second sentence you may move to the end of the introduction or only have in the background.
Maybe it will be better to move line 74-77, after the 1.2. Theoretical perspective. The aim is closed to the method section.
Please include reference 19 in the last sentence in line 95.
It is only reference 19 in the heading 1.2. Theoretical perspective? Or have you more references? In other wise, please include the reference (19) after every section or write together.
Method:
A question on your manual searches, why these specific journals? Other journals?
Do you search for grey literature?
Who performed the primary searches? All authors?
It is Table 2 more than Paper Characteristics? Please specify
Table 2. Included studies in the meta…. And information about …
Results:
My recommendation is to start the results with line 199.
Line 196-198 don’t contribute with a good start of the result section. If you will introduce with the subheadings, please clarify what you mean, not clear in this way. Introduce the metaphor and the subheadings in the beginning of the result section.
How you relate your results to the theoretical framework? I don’t see the read thread related to your background.
Discussion:
The start of your discussion is good, and you highlight the main results/findings.
My recommendation is to move the text about Eriksson (line 472), early in the discussion part and relevant to your aim of the synthesis.
What your strengths of your study?
The line 490 – 500 is very general and not specific.
My recommendation is to also improve the limitations of your study. It is very sparse.
Tables and figures:
Overall good and easy to read and understand.
Author Response
Dear reviewer 2,
Thank you for your thoughts on the manuscript ijerph-704190_Involuntary pregnancy loss and nursing care: A meta-ethnography. In the following table we detail the answers and modifications made in the manuscript (See attached document).

Reviewer 3 Report
The current paper presents a meta-ethnography review of 10 qualitative studies to synthesize nurses’ and midwives’ experiences of caring for parents following an involuntary pregnancy loss. The study helps understand the concept of ‘caring in darkness’ and the complexity of experiences of nursing and midwifery in caring for these parents. The study is of high quality as authors have taken a meticulous and rigorous approach to each aspect of the study: design, methodology, analysis and discussion. The authors had conducted a comprehensive literature search and the study selection process was systematic. The quality of each eligible study was assessed and inclusion and exclusion criteria and the reasons for including final 10 studies are clearly described. Characteristics of the included studies have been presented in the tables. Discussion includes the strengths, limitations and relevance of the study findings to perinatal clinical practice in detail.
I have few comments to make:
Some of the sentences are not easily comprehensible. Restructuring or simplifying those sentences would make them an easy read. For e.g. Page 1, line 45 “ The intrinsic feature…”; Page 2, line 57 “ The emotional signs…”; Page 2, line 59 “The need to support…” Page 2, line 53. Not sure the word ‘relief’ is appropriate Page 2, line 47, ‘the estimated 2.6 million babies..’ needs a reference and clarification, if this incidence rate is for the entire world. If the quality assessment of the articles done by any independent evaluator. If the authors looked at the differential experiences of the nurses and midwives caring for those parents who suffered first time pregnancy loss vs. multiple pregnancy losses. Search terms didn’t appear to include ‘experience’ or ‘care’. Generalizability is limited : Studies included are from few countries/western cultures only Lack of studies regarding experiences of midwives working in the community/primary care settings Lack of studies regarding experiences of working with teenager parents with pregnancy losses.Author Response
Dear reviewer 3,
Thank you for your insights and comments to the manuscript ijerph-704190_Involuntary pregnancy loss and nursing care: A meta-ethnography. The following table details our answers (see attached document).
